# *Drosophila* neprilysins control insulin signaling and food intake via cleavage of regulatory peptides

Benjamin Hallier[1†], Ronja Schiemann[1†], Eva Cordes[1], Jessica Vitos-Faleato[2], Stefan Walter[3], Jürgen J Heinisch[4], Anders Malmendal[5], Achim Paululat[1], Heiko Meyer[1*]

[1]Department of Developmental Biology, University of Osnabrück, Osnabrück, Germany; [2]Department of Biomedical Research, Institute for Research in Biomedicine, Barcelona, Spain; [3]Department of Microbiology, University of Osnabrück, Osnabrück, Germany; [4]Department of Genetics, University of Osnabrück, Osnabrück, Germany; [5]Department of Cellular and Molecular Medicine, University of Copenhagen, Copenhagen, Denmark

*For correspondence: Meyer@ biologie.uni-osnabrueck.de

[†]These authors contributed equally to this work

Competing interests: The authors declare that no competing interests exist.

**Abstract** Insulin and IGF signaling are critical to numerous developmental and physiological processes, with perturbations being pathognomonic of various diseases, including diabetes. Although the functional roles of the respective signaling pathways have been extensively studied, the control of insulin production and release is only partially understood. Herein, we show that in *Drosophila* expression of insulin-like peptides is regulated by neprilysin activity. Concomitant phenotypes of altered neprilysin expression included impaired food intake, reduced body size, and characteristic changes in the metabolite composition. Ectopic expression of a catalytically inactive mutant did not elicit any of the phenotypes, which confirms abnormal peptide hydrolysis as a causative factor. A screen for corresponding substrates of the neprilysin identified distinct peptides that regulate insulin-like peptide expression, feeding behavior, or both. The high functional conservation of neprilysins and their substrates renders the characterized principles applicable to numerous species, including higher eukaryotes and humans.

## Introduction

Neprilysins are highly conserved ectoenzymes that cleave and thereby inactivate many physiologically relevant peptides in the extracellular space, thus contributing considerably to the maintenance of peptide homeostasis in this compartment. Members of the neprilysin family generally consist of a short N-terminal cytoplasmic domain, a membrane spanning region, and a large extracellular domain with two highly conserved sequence motifs (HExxH; ExxA/GD) critical for zinc coordination, catalysis, and substrate or inhibitor binding (*Matthews, 1988*; *Oefner et al., 2000*). Because of these characteristics, neprilysins are classified as M13 zinc metallopeptidases. For human Neprilysin (NEP), the most well-characterized family member, identified substrates include endothelins, angiotensins I and II, enkephalins, bradykinin, atrial natriuretic peptide, substance P, and the amyloid-beta peptide (*Turner et al., 2001*). Because of this high substrate variability, NEP activity has been implicated in the pathogenesis of hypertension (*Molinaro et al., 2002*), analgesia (*Whitworth, 2003*), cancer (*Turner et al., 2001*), and Alzheimer's disease (*Iwata et al., 2000*; *Belyaev et al., 2009*). Recent clinical trials have demonstrated significant efficacy of Neprilysin inhibitors in the treatment of certain indications (*Jessup, 2014*; *McMurray et al., 2014*). However, despite the clinical relevance of the neprilysins, the physiological function and *in vivo* substrates of most family members are unknown.

**eLife digest** The hormone insulin and similar molecules called insulin-like peptides act as signals to control many processes in the body, including growth, stress responses and aging. Disrupting these signaling pathways can cause many diseases, with diabetes being the most common of these. Although the roles of the signaling pathways have been well studied, it is less clear how the body controls the production of insulin and insulin-like peptides.

Neprilysins are enzymes that can cut other proteins and peptides by a process known as hydrolysis. Their targets (known as "substrates") include peptides that regulate a range of cell processes, and neprilysins have therefore been linked with many diseases. Fruit flies have at least five different neprilysin enzymes, but their substrates have not yet been identified. One of these, known as Nep4A, is produced in muscle tissue and appears to be important for muscles to work properly.

Hallier, Schiemann et al. reveal that Nep4A regulates the production of insulin-like peptides. The experiments used fruit fly larvae that had been genetically engineered so that the level of Nep4A could be altered in muscle tissue. Larvae with very high or very low levels of Nep4A eat less food, have smaller bodies and produce different amounts of insulin-like peptides compared to normal larvae.

Further experiments show that Nep4A can hydrolyze a number of peptides that regulate the production and the release of insulin-like peptides. This suggests that the enzymatic activity of neprilysins plays a direct role in controlling the production of insulin. The next challenge is to find out whether these findings apply to humans and other animals that also have neprilysins.

In *Drosophila melanogaster*, at least five neprilysin genes are expressed (*Meyer et al., 2011*; *Sitnik et al., 2014*), two of the corresponding protein products, Nep2 and Nep4, were reported to be enzymatically active (*Bland et al., 2007*; *Meyer et al., 2009*; *Thomas et al., 2005*). With respect to Nep4, a critical function of the enzyme's non-catalytic intracellular N-terminus has been demonstrated: when present in excess, the domain induces severe muscle degeneration concomitant with lethality during late larval development. Because the intracellular domain interacts with a carbohydrate kinase, impaired energy metabolism has been proposed as the underlying cause of the phenotype (*Panz et al., 2012*). In addition, Nep2 has been implicated in the regulation of locomotion and geotactic behavior (*Bland et al., 2009*), and neprilysin activity in general appears to be critical to the formation of middle- and long-term memory (*Turrel et al., 2016*), as well as to the regulation of pigment dispersing factor (PDF) signaling within circadian neural circuits (*Isaac et al., 2007*). However, despite these experiments and recent findings that suggest a critical role of neprilysins in reproduction (*Sitnik et al., 2014*), the physiological functionality of these enzymes is still far from being understood. In this respect, the lack of identified substrates with *in vivo* relevance is a major hindrance.

Herein, we describe the identification of numerous novel substrates of *Drosophila* Neprilysin 4 (Nep4) and provide evidence that Nep4-mediated peptide hydrolysis regulates insulin-like peptide (ILP) expression and food intake. These results establish a correlation between neprilysin activity and ILP expression and thus clarify our understanding of the complex mechanisms that control the production and release of these essential peptides.

## Results

### Modulating the expression of Neprilysin 4 affects lifespan and body size

In previous experiments, we showed that Nep4 is expressed in larval body wall muscles and that increased expression of the peptidase in this tissue interferes with muscle function and integrity and severely impairs movement of the larvae (*Panz et al., 2012*). In the present study, we found that an increase in Nep4A in muscle cells (*mef2*-Gal4 driver) also induced biphasic lethality. An early phase occurred throughout embryonic and early larval development, and a late phase was evident by the

end of larval development (*Figure 1A*). Significantly, early lethality was observed only upon overexpression of the active enzyme; expression of catalytically inactive Nep4A, carrying a glutamine instead of an essential glutamate (E873Q) within the zinc-binding motif, did not affect viability at this point of development. By contrast, overexpression of catalytically active or inactive Nep4A constructs induced late larval lethality. These distinct effects demonstrate that lethality during early development is caused exclusively by a detrimental increase in catalytic activity, whereas late larval lethality appears to be a consequence of multiple physiological impairments. Comparable overexpression levels of the wild-type enzyme and the mutated construct were demonstrated previously (*Panz et al., 2012*). Muscle-specific knockdown of *nep4* slightly increased embryonic mortality, but the majority of the respective animals died during metamorphosis (*Figure 1A*). To confirm RNAi specificity, we also analyzed flies expressing both the respective RNAi construct as well as the Nep4A overexpression construct. Simultaneous overexpression of Nep4A completely rescued the RNAi phenotypes (embryonic/pupal lethality), thus confirming specificity of the knockdown (*Figure 1A*). The result that respective animals exhibited a marginally, yet significantly increased lethality rate during third instar larval stage indicates that overexpression of Nep4A is somewhat more effective than knockdown, eventually resulting in slightly increased expression levels of the peptidase, which, as depicted above, result in elevated larval lethality.

As shown previously, in addition to muscle tissue *mef2* is expressed in distinct neurons, including clock neurons (*Blanchard, 2010*) and Kenyon cells (*Schulz et al., 1996*). To determine whether the effects described above (using *mef2*-Gal4 as a driver) are exclusively based on Nep4 activity in muscles, or if neuronal Nep4 is also involved, we used pan-neuronal *elav*-Gal4 as a driver to increase or reduce *nep4* expression. In this line of experiments, neither overexpression nor knockdown of *nep4* had any significant influence on viability (*Figure 1A*). This result indicates that the effects observed with *mef2*-Gal4 are muscle-specific.

In addition to muscle tissue, Nep4 is also expressed in glial cells of the central nervous system (CNS) (*Meyer et al., 2009*). However, in contrast to the effects observed in muscle cells, neither increased nor reduced *nep4* expression in glial cells, using glia-specific *repo*-Gal4 as a driver, significantly affected life span (*Figure 1A*).

Besides reduced viability, elevated Nep4A levels in muscle tissue affected body size. Interestingly, as with increased lethality during early development, the effects on body size depended on enzymatic activity. In third instar larvae, muscle-specific overexpression of the active enzyme decreased the size and weight of the animals relative to control animals, whereas overexpression of catalytically inactive Nep4A did not affect size or weight. Knockdown of the peptidase in the same tissue did also not significantly alter these parameters (*Figure 1B*). In contrast to the muscle-specific effects, increased *nep4A* expression in glial cells or neurons did not affect the size or weight of the larvae. Glial cell specific *nep4* knockdown slightly reduced both parameters, whereas neuronal knockdown had no effect (*Figure 1B*). In line with the lethality assay, the effects of *nep4* knockdown on size and weight were completely rescued by simultaneous overexpression of Nep4A, which again confirms specificity of the respective RNAi construct. The depicted results indicate essential functions of Nep4 in muscle tissue and glial cells. However, the effects of modifying *nep4* expression were more severe in muscles, suggesting the active enzyme has a critical function particularly in this tissue.

## Modulating the expression of Neprilysin 4 interferes with basal metabolic processes

To understand the physiological basis of this function in more detail, we analyzed the metabolite composition in animals with increased or reduced *nep4* levels and compared the respective compositions to those in control specimens. As shown in *Figure 2A*, increasing or decreasing the expression of *nep4* in muscle tissue affected metabolite concentrations in transgenic third instar larvae. Of note, the depicted PCA scores plot is purely based on the amplitude of correlated between-sample variations, implicating that the strongest variations in metabolite composition are those separating the three genotypes. Further analysis of the respective data revealed that profound changes are related to the energy metabolism. Knockdown of *nep4* increased the levels of fructose and a purine and decreased the levels of NAD, a purine nucleotide, and glutamine (*Figure 2B,C*, *Figure 2—source data 1*). Nep4A overexpression increased the signals of histidine, glutamine, and the same purine. In addition, a significant increase was observed in the spectral regions specific to glucose and fructose, indicating elevated levels of the two monosaccharides. Of note, only the glucose and

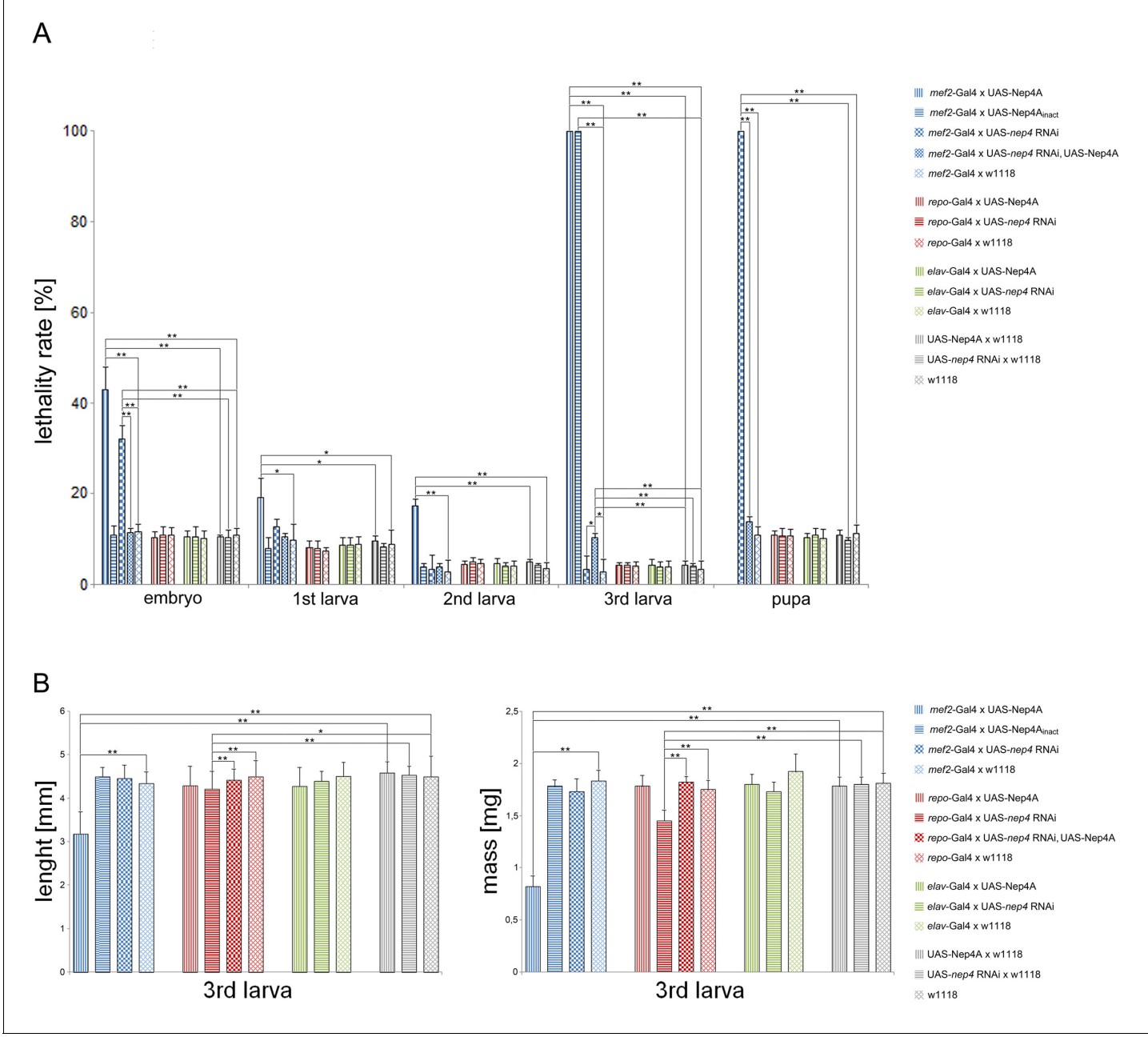

**Figure 1.** Modulating *nep4* expression affects life span and body size. (**A**) Lethality assay. The percentages (%) of animals of a specific stage that did not develop into the next stage are shown. While muscle-specific overexpression of Nep4A (*mef2*-Gal4 x UAS-Nep4A) led to biphasic lethality with critical phases during embryonic and late larval development, overexpression of catalytically inactive Nep4A in the same tissue (*mef2*-Gal4 x UAS-Nep4A_inact) led to lethality only in the third instar larval stage. Muscle-specific *nep4* knockdown (*mef2*-Gal4 x UAS-*nep4* RNAi) slightly increased embryonic lethality, but the majority of the animals died as pupae. Glial cell-specific overexpression (*repo*-Gal4 x UAS-Nep4A) or knockdown of the peptidase (*repo*-Gal4 x UAS-*nep4* RNAi) did not affect life span, which was also observed for neuronal overexpression or knockdown (*elav*-Gal4 x UAS-Nep4A; *elav*-Gal4 x UAS-*nep4* RNAi). *mef2*-Gal4 x w1118, *repo*-Gal4 x w1118, *elav*-Gal4 x w1118, UAS-Nep4A x w1118, UAS-*nep4* RNAi x w1118, and w1118 were used as controls. Asterisks indicate statistically significant deviations from the respective controls (*p<0.05, **p<0.01, one-way ANOVA with pairwise comparisons). (**B**) Size and weight measurements. While muscle-specific overexpression of Nep4A (*mef2*-Gal4 x UAS-Nep4A) reduced the size and wet mass of third instar larvae, neither overexpression of catalytically inactive Nep4A in the same tissue (*mef2*-Gal4 x UAS-Nep4A_inact) nor muscle-specific *nep4* knockdown (*mef2*-Gal4 x UAS-*nep4* RNAi) significantly affected these parameters. Glial cell-specific overexpression of the peptidase (*repo*-Gal4 x UAS-Nep4A) did not alter size or weight, while downregulation of the peptidase in the same tissue (*repo*-Gal4 x UAS-*nep4* RNAi) slightly, but significantly, reduced both parameters. Neuronal overexpression or knockdown of *nep4* (*elav*-Gal4 x UAS-Nep4A; *elav*-Gal4 x UAS-*nep4* RNAi) had

*Figure 1 continued on next page*

*Figure 1 continued*

no effect on size or weight. Control lines were the same as in A. Asterisks indicate statistically significant deviations from respective controls (*p<0.05, **p<0.01, one-way ANOVA with pairwise comparisons).

The following source data is available for figure 1:

**Source data 1.** Lethality assay.
**Source data 2.** Size and weight measurements.

fructose signals with contributions from both sugars (depicted in *Figure 2B*) were significantly affected, implying that there is a more stable response in the sum of the two than in either of them. However, evaluation of the corresponding individual spectra clearly suggested that both sugars are increased (*Figure 2—figure supplement 1*). On the other hand, lactate, NAD, trehalose, and tyrosine concentrations were reduced in Nep4A-overexpressing animals (*Figure 2B,C*, *Figure 2—source data 1*). Of note, increased formation of lactate and NAD is a hallmark of aerobic glycolysis, a specific metabolic program that starts approximately 12 hr before the end of embryogenesis. Aerobic glycolysis enables hatched 1st instar larvae to efficiently convert dietary carbohydrates into biomass, thereby supporting the considerable increase in body mass that occurs during larval development (*Tennessen et al., 2014*). Inhibition of aerobic glycolysis in the course of this growth phase prevents the animals from metabolizing sufficient quantities of sugar, resulting in larval lethality (*Tennessen et al., 2011*). The fact that animals overexpressing Nep4A die primarily during the embryonic-larval transition and during larval development (*Figure 1A*) and exhibit considerably reduced lactate and NAD levels indicates that an excess of Nep4A may interfere with this distinct metabolic program. OPLS-DA loading plots summarizing the respective NMR spectral changes are depicted in *Figure 2C*.

## Neprilysin 4 activity regulates food intake and insulin-like peptide expression

Given that the described metabolic abnormalities are indicative of an impaired energy metabolism, we analyzed whether modulating *nep4* expression affects feeding of corresponding animals. As depicted in *Figure 3A*, transgenes overexpressing the peptidase were characterized by considerably reduced food intake. After 10 min of feeding, the respective animals had ingested 47% less food than controls, after 20 min 59.5% less, and after 40 min 57% less, relative to controls. By contrast, *nep4* knockdown did not affect food intake after 40 min; however, corresponding animals were characterized by significantly reduced food intake after 10 min (47% of control intake) and 20 min (72% of control intake), indicating a delayed initiation of feeding. To investigate the possibility that the observed effects were caused by protein properties other than enzymatic activity, we also analyzed catalytically inactive Nep4A. Significantly, overexpression of this construct did not affect food intake, thus confirming abnormal catalytic activity as a causative factor (*Figure 3A*).

Since the increased glucose levels that are evident in Nep4A overexpression animals (*Figure 2B, C*) are symptomatic of impaired insulin signaling (*Broughton et al., 2005*; *Rulifson et al., 2002*), in a continuative set of experiments we analyzed whether altering *nep4* levels also affected the expression of *Drosophila* insulin-like peptides (*dilps*). We focused on *dilps 1*, *2*, *3*, and *5* because they encode the major insulin-like peptides expressed by larval insulin-producing cells (IPCs) (*Rulifson et al., 2002*; *Brogiolo et al., 2001*; *Cao and Brown, 2001*; *Ikeya et al., 2002*; *Lee et al., 2008*; *Nässel et al., 2013*). IPCs are located within the median neurosecretory cell cluster of the central brain and apparently function like pancreatic β-cells, since IPC ablation in *Drosophila* results in elevated levels of circulating glucose. In addition, animals with ablated IPCs are smaller than wild-type specimens, and they weigh less (*Broughton et al., 2005*; *Rulifson et al., 2002*). Significantly, these characteristic effects of IPC ablation were phenocopied by muscle-specific Nep4A overexpression (*Figures 1* and *2*). Furthermore, the respective transgenic animals exhibited considerably reduced expression of the selected *dilps*. In Nep4A-overexpressing animals, *dilp1* expression decreased by 59%, *dilp2* by 83%, *dilp3* expression by 88%, and *dilp5* expression by 84%, relative to expression in controls. On the other hand, muscle-specific *nep4* knockdown had no effect on the

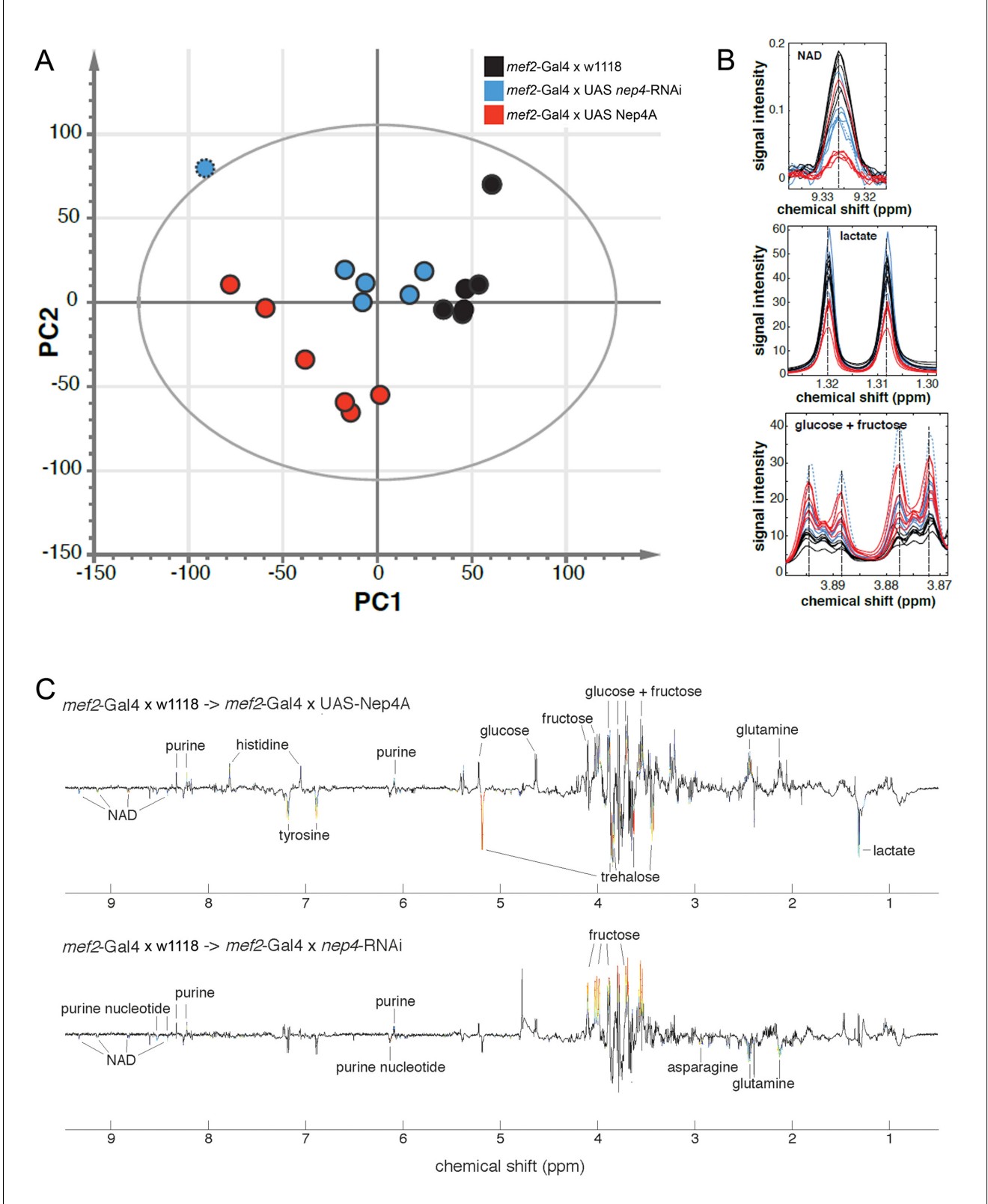

**Figure 2.** Muscle-specific modulation of *nep4* expression affects the metabolite composition in transgenic third instar larvae. (A) Score plot based on genotype-specific NMR spectra. PCA score plot showing the scores of six biological replicates for each genotype. Principal component analysis (PCA) was applied to identify metabolite changes in response to muscle-specific Nep4A overexpression (*mef2*-Gal4 x UAS-Nep4A; red) or knockdown (*mef2*-Gal4 x UAS *nep4*-RNAi; blue), relative to control animals (*mef2*-Gal4 x w1118; black). The score plot reveals genotype-specific clustering and thus

*Figure 2 continued on next page*

*Figure 2 continued*

distinct metabolite compositions in corresponding animals. One *nep4* knockdown sample was distinctly different from the other five. The outlier is marked by a dotted border and was excluded from OPLS-DA identification of significantly affected metabolites. (B) Examples of NMR signals from significantly affected metabolites. Evaluation of the dataset revealed that Nep4A overexpression significantly reduced NAD and lactate concentrations, while glucose and fructose levels were elevated in the same animals. The effects of *nep4* knockdown were less severe; NAD was reduced, and fructose was slightly elevated, compared to levels in control animals. The coloring is the same as in A. The knockdown outlier is marked by a dotted line. (C) OPLS-DA loading plots summarizing the NMR spectral changes induced by *nep4* overexpression and knockdown. Depicted is an overview of the metabolomic changes induced by modifying the expression of *nep4*. Positive and negative signals represent increases and decreases in metabolite concentrations, respectively. Significant alterations are color-coded from blue to red. Red represents the highest correlation between metabolite and genotype.

The following source data and figure supplement are available for figure 2:

**Source data 1.** Chemical shifts and detected changes of significantly affected metabolites.
**Figure supplement 1.** NMR-spectra of glucose and fructose.

expression of *dilps 1, 3,* and *5,* although expression of *dilp2* increased by 82%, relative to expression in controls (*Figure 3B*). The rather mild effect of *nep4* knockdown on *dilp* expression, when compared to the effects of Nep4A overexpression, suggests that other, yet unknown peptidases can compensate for reduced Nep4 activity. In line with the results from the feeding assay (*Figure 3A*), *dilp* expression is only affected by the wild type enzyme, while overexpression of catalytically inactive Nep4A did not significantly alter expression of the selected *dilps* (*Figure 3B*).

To determine if expression of insulin-like peptides is regulated exclusively by muscle-derived Nep4 or if intrinsic CNS signaling is also involved, we altered *nep4* expression in a nervous-system-specific manner. As depicted in *Figure 3—figure supplement 1*, glial-cell-specific overexpression of Nep4A increased the expression of *dilp5*, while *nep4* knockdown in the same cells resulted in an upregulation of *dilp2* and downregulation of *dilp3*. Although these effects were minor compared to the effects of modulating the expression of muscle-bound Nep4 (*Figure 3B*), they demonstrate that proper regulation of *dilp* expression also requires adequate Nep4 levels within the CNS.

## Neprilysin 4 localizes to the surface of larval body wall muscles and IPCs

In order to understand the physiological relation between *dilp* expression and Nep4 activity in more detail, we analyzed the expression pattern and the subcellular localization of the peptidase in larval body wall muscles and the larval CNS. As depicted in *Figure 4*, in body wall muscles Nep4 exhibits a dual localization: in addition to localizing to membranes continuous with the nuclear membrane (*Figure 4A*, arrowheads), which we previously identified as related to the sarco/endoplasmic reticulum (*Panz et al., 2012*), the peptidase accumulates at the surface of the muscles (*Figure 4A*, arrows). The latter localization is consistent with ectoenzymatic activity and indicative of a function in regulating the homeostasis of hemolymph circulating peptides. To confirm the specificity of the signal, we also stained the muscles of transgenic animals expressing *nep4*-specific RNAi (*mef2*-Gal4 driver). In these transgenic animals, no signal above background was observed (*Figure 4B*). In addition, staining wild-type muscles with secondary antibodies alone did not result in a distinct signal (*Figure 4C*). Of note, the proteins expressed from the two overexpression constructs (wild-type Nep4A and catalytically inactive Nep4A) exhibited subcellular localizations identical to that of the endogenous protein (*Figure 4A,D,F*), confirming that the observed overexpression phenotypes (*Figures 1–3*) were not impaired by mislocalization of the respective constructs. In order to distinguish the ectopic proteins from the endogenous protein, the ectopic constructs were fused to a C-terminal HA-tag and labeled with corresponding antibodies. Antibody specificity was confirmed by the lack of staining in animals expressing only the Gal4 transgene but not the UAS-construct (*Figure 4E,G*).

To characterize expression in the CNS, we employed a reporter line that expresses nuclear GFP (nGFP) in a manner that recapitulates endogenous *nep4* expression (*Meyer et al., 2009*). As shown in *Figure 5*, brain and ventral nerve cord tissue exhibited substantial reporter gene expression. With respect to the brain, expression was observed mainly in lamina (*Figure 5A*, brackets) and central

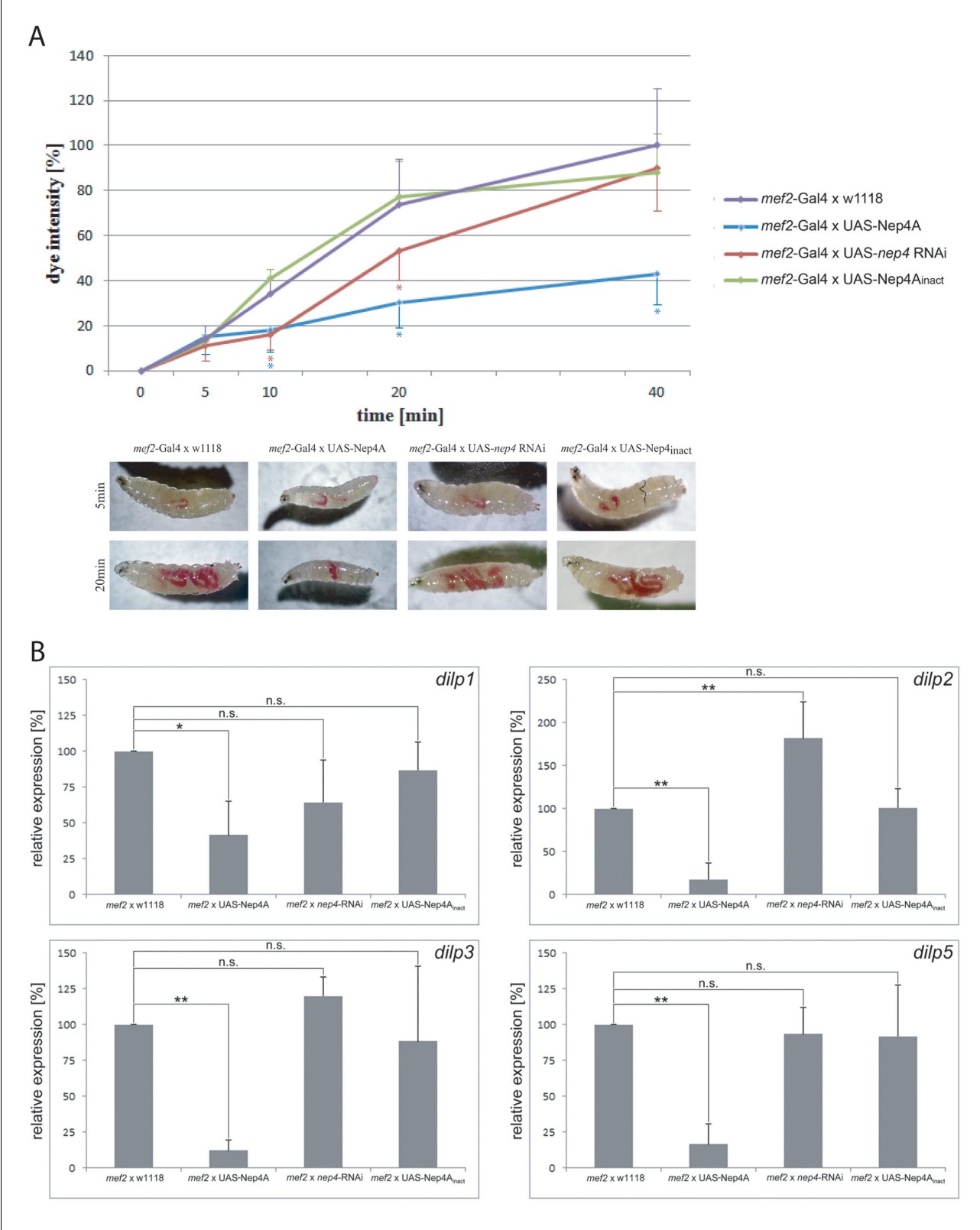

**Figure 3.** Muscle-specific modulation of *nep4* expression affects food intake and *dilp* expression in transgenic third instar larvae. (**A**) The genotype-specific rates of food intake are depicted as percentages (%) relative to the intake in control specimens (*mef2*-Gal4 x w1118) after 40 min of feeding, which was set to 100%. While *nep4* knockdown animals (*mef2*-Gal4 x UAS-*nep4* RNAi) exhibited reduced food intake after 10 and 20 min of feeding, Nep4A overexpression animals (*mef2*-Gal4 x UAS-Nep4A) were characterized by reduced intake throughout the whole measurement (up to 40 min). *Figure 3 continued on next page*

*Figure 3 continued*

Animals overexpressing catalytically inactive Nep4A (*mef2*-Gal4 x UAS-Nep4A$_{inact}$) did not exhibit any significant changes in food intake, when compared to controls. Values represent the mean (± s.d.) of at least six independent biological replicates. Asterisks indicate statistically significant deviations from controls (*p<0.05, one-way ANOVA with pairwise comparisons). The lower panel depicts representative images of the genotype-specific food intake at the indicated time points. (B) Changes in the expression of selected *dilp* genes are presented as percentages (%) relative to expression in control specimens (*mef2*-Gal4 x w1118), which was set to 100%. Muscle-specific overexpression of Nep4A (*mef2* x UAS-Nep4A) reduced the expression of every *dilp* gene analyzed, while *nep4* knockdown in the same tissue (*mef2* x *nep4*-RNAi) resulted in upregulation of *dilp2*. Animals overexpressing catalytically inactive Nep4A (*mef2* x UAS-Nep4A$_{inact}$) did not exhibit any significant changes in *dilp* expression, when compared to controls. Values represent the mean (+ s.d.) of at least three independent biological replicates, each consisting of at least three technical replicates. Asterisks indicate statistical significance (*p<0.1; **p<0.05, one-way ANOVA with pairwise comparisons); n.s. indicates 'not significant'.

The following source data and figure supplement are available for figure 3:

**Source data 1.** Feeding assay.
**Figure supplement 1.** Glial cell-specific modulation of *nep4* expression affects *dilp* expression in transgenic third instar larvae.

brain cells (*Figure 5A*, dashed line), while only a few medulla cells exhibited a distinct signal (*Figure 5A*, bar). Within the ventral nerve cord, *nep4* was detected in numerous cells along all segments. As confirmed by extensive colocalization with the glial cell marker Reversed-polarity (Repo), *nep4* was expressed primarily in this cell type; however, especially in the median region of the central brain, only partial colocalization was evident. Thus, in addition to glial cells, *nep4* is expressed in certain neurons of the central brain (*Figure 5A-C*).

Interestingly, as confirmed by colocalization with *dilp2*-specific reporter gene expression, we found that these neurons included all IPCs, which reside within the median neurosecretory cell

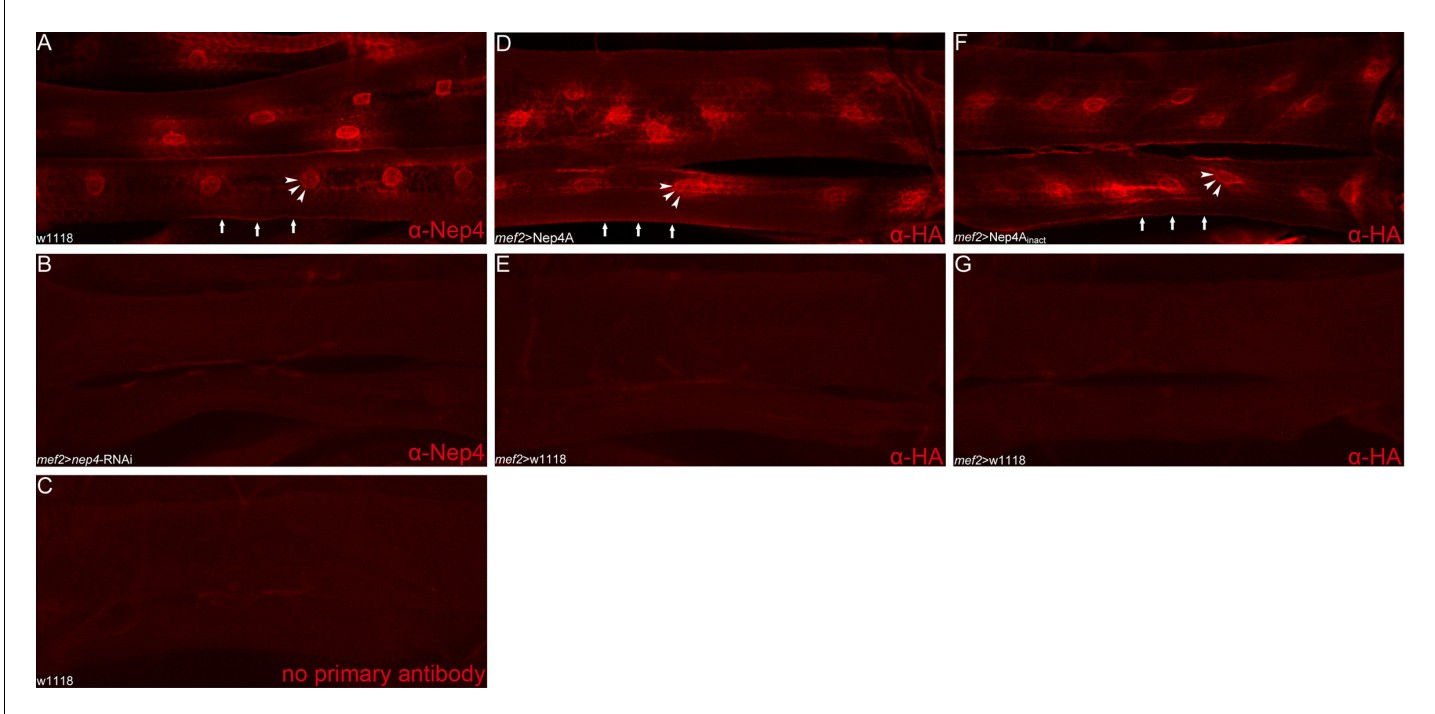

**Figure 4.** Nep4 localizes to the surface of muscle cells. (A) Nep4 protein was labeled with a monospecific antibody (red). In addition to membranes continuous with the nuclear membrane (arrowheads), Nep4 accumulated at the surface of body wall muscles (arrows). (D, F) Nep4 overexpression constructs (*mef2*>Nep4A, *mef2*>Nep4A$_{inact}$) exhibited subcellular localizations identical to that of the endogenous protein. The corresponding constructs were labeled with antibodies detecting the fused HA-tag. (B, C, E, G) Control stainings did not produce any signal above background.

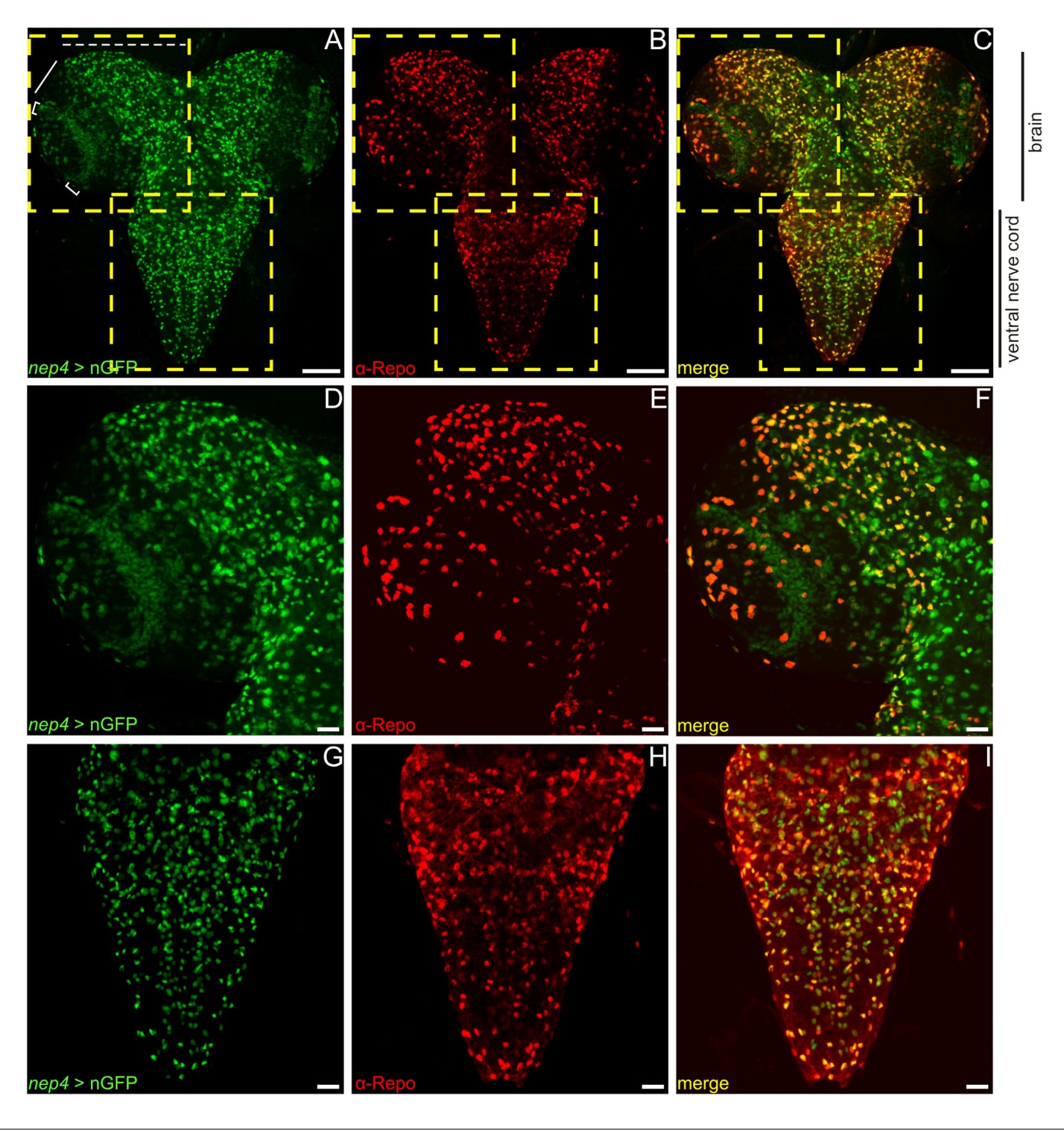

**Figure 5.** Nep4 is expressed in glial cells and neurons in the central nervous system. *nep4* expression was visualized using a reporter construct that drives nuclear GFP (nGFP) expression in a *nep4*-specific manner (*nep4* > nGFP, green). Reversed polarity protein was labeled with a monospecific antibody (α-Repo, red). (**A–C**) Optical projections of third instar larval whole brain-ventral nerve cord complexes. Scale bars: 100 µm; dorsal view, anterior up. Boxes indicate areas of higher magnification, as depicted in (**D–F**) and (**G–I**). Within the brain, *nep4* expression was strongest in the central brain (**A**, dashed line) and in lamina cells (**A**, brackets), while only few *nep4*-positive medulla cells were observed (**A**, bar). Within the ventral nerve cord,

*Figure 5 continued on next page*

Hallier *et al.* eLife 2016;5:e19430. DOI: 10.7554/eLife.19430

*Figure 5 continued*

*nep4* was expressed in numerous cells along all segments. (**D–I**) Optical projections of third instar larval brain hemisphere (**D–F**) and ventral nerve cord (**G–I**). Scale bars: 20 µm; dorsal view, anterior up, midline to right. *nep4* expression colocalized extensively with anti-Repo staining.

cluster of the brain hemispheres. The distinct localization of the respective signals is because both reporter constructs drive expression of a nuclear localized fluorophore (**Figure 6A–C**). To assess the subcellular localization of Nep4 in IPCs, we performed double labeling experiments using a reporter line expressing eGFP in a *dilp2*-specific manner, thus labeling the IPC cytoplasm, together with Nep4-specific antibodies. As depicted in **Figure 6D–F**, the peptidase accumulated at the surface of numerous cells of the central brain, including IPCs.

## Neprilysin 4 efficiently hydrolyzes peptides that regulate *dilp* expression and feeding behavior

The fact that major phenotypes described in this study strictly depend on the catalytic activity of Nep4 (**Figures 1** and **3**) indicates that aberrant hydrolysis of peptides involved in regulating *dilp*

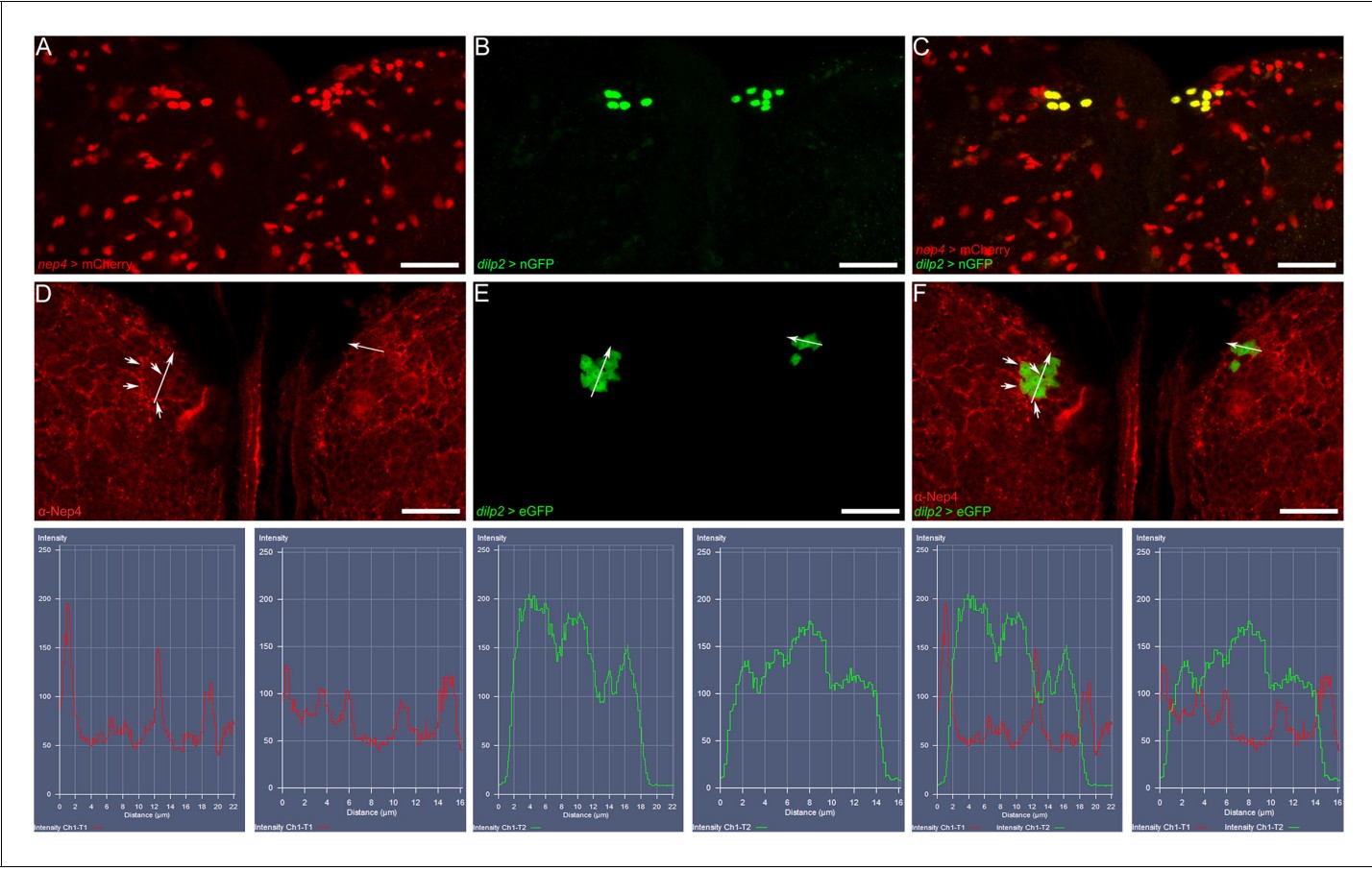

**Figure 6.** Nep4 localizes to the surface of insulin-producing cells. (**A–C**) *nep4* expression was assessed using a reporter line that drives nuclear mCherry expression in a *nep4*-specific manner (*nep4* > mCherry, red). *dilp2* expression was visualized using a reporter construct that drives nuclear GFP expression in a *dilp2*-specific manner (*dilp2* > nGFP, green). Depicted are optical sections (10 µm) of a third instar larval central brain. Scale bars: 20 µm; dorsal view, anterior up. *nep4* and *dilp2* expression colocalized in IPCs. (**D–F**) Nep4 protein was labeled with a monospecific antibody (red), and *dilp2* expression was visualized using an eGFP reporter line (*dilp2* > eGFP, green). Depicted are optical sections (10 µm) of a third instar larval central brain. Scale bars: 20 µm; dorsal view, anterior up. Nep4 accumulated at the surface of numerous cells, including IPCs (**D**, **F**, arrowheads). The subcellular localization was assessed with fluorescence intensity measurements (lower panel). The respective regions of evaluation are marked (arrows in **D–F**).

expression and / or feeding behavior is primarily responsible for the phenotypes. This indication is corroborated by the localization of Nep4 to the surface of body wall muscles and IPCs (*Figures 4* and *6*), with the latter constituting the major site of Dilp synthesis in *Drosophila* (*Rulifson et al., 2002*; *Brogiolo et al., 2001*; *Ikeya et al., 2002*). In order to identify the causative hydrolysis event (s), we analyzed every peptide known to be involved in regulating *dilp* expression, feeding behavior, or both (*Nässel et al., 2013*; *Pool and Scott, 2014*) for susceptibility to Nep4-mediated cleavage. The only additional prerequisite for consideration as a potential substrate was a size of less than 5 kDa, which for steric reasons represents the maximum mass of a neprilysin substrate (*Oefner et al., 2000*). As depicted in *Table 1*, we found 23 peptides matching these criteria. Among these candidates, 16 were hydrolyzed at distinct positions by purified Nep4, while the remaining seven peptides were not significantly cleaved. The identified substrates were adipokinetic hormone (AKH), allostatin A1-4, corazonin, diuretic hormone 31 (DH$_{31}$), drosulfakinins 1 and 2, leucokinin, short neuropeptide F1$_{1-11}$, short neuropeptide F1$_{4-11}$ (also corresponding to sNPF2$_{12-19}$), and tachykinins 1, 2, 4, and 5. No Nep4-specific cleavage was observed for hugin, neuropeptide F, proctolin, short neuropeptide F3, short neuropeptide F4, and tachykinins 3 and 6. Analysis of the resulting hydrolysis products revealed that Nep4 preferentially cleaved next to hydrophobic residues, particularly with Phe or Leu at P1′ (*Table 1*). Identically treated control preparations lacking the peptidase did not exhibit any cleavage activity (*Figure 7*). Individual MS chromatograms are depicted in *Figure 7*. Nep4B purity was confirmed with SDS-PAGE (*Figure 7—figure supplement 1*).

## Discussion

While the functional roles of insulin-like peptides (ILPs) and the corresponding insulin- and IGF-signaling have been intensively studied, the control of ILP production and release is not well understood. This study demonstrates that modulating the expression of a *Drosophila* neprilysin interferes with the expression of insulin-like peptides, thus establishing a correlation between neprilysin activity and the regulation of insulin signaling. A high physiological relevance is confirmed by the fact that altering *nep4* expression phenocopies characteristic effects of IPC ablation, including reduced size and weight of corresponding animals, as well as increased levels of carbohydrates such as glucose and fructose (*Figures 1* and *2*). The result that the levels of these sugars are increased, although food intake rates are reduced (*Figure 3A*) presumably reflects the physiological impact of the diminished *ilp* expression that is also obvious in corresponding animals (*Figure 3B*). In this respect, the impaired insulin signaling likely results in inefficient metabolization and thus accumulation of the sugars, which overcompensates the diametrical effects of reduced food intake. By identifying 16 novel peptide substrates of Nep4, the majority of which are involved in regulating *dilp* expression or feeding behavior (*Table 1*, *Figure 7*), and by localizing the peptidase to the surface of body wall muscles (*Figure 4*) and IPCs within the larval CNS (*Figure 6*), we provide initial evidence that neprilysin-mediated hydrolysis of hemolymph circulating as well as CNS intrinsic peptides is the physiological basis of the described phenotypes. The finding that only the catalytically active enzyme affected *dilp* expression whereas the inactive construct did not (*Figure 3B*), substantiates this evidence because it confirms aberrant enzymatic activity and thus abnormal peptide hydrolysis as a causative parameter. Interestingly, we observed the strongest effects on size and *dilp* expression with muscle-specific overexpression of Nep4; overexpression of the peptidase in the CNS was less detrimental (*Figure 1*, *Figure 3B*, *Figure 3—figure supplement 1*). These results indicate that hemolymph circulating peptides accessible to muscle-bound Nep4 are mainly responsible for the observed effects, while CNS intrinsic peptide signaling is less relevant. The fact that all peptides cleaved by Nep4 (*Table 1*) could be released into the hemolymph, either from enteroendocrine cells or from neurohormonal release sites (*Nässel and Winther, 2010*), substantiates this indication. Since the *Drosophila* midgut is the source of several neuropeptides (*Veenstra et al., 2008*; *Reiher et al., 2011*), it is conceivable that a main reason for the observed phenotypes is aberrant cleavage of certain gut-derived peptides that are required for proper midgut-IPC communication. Allatostatin A, neuropeptide F, diuretic hormone 31, and some tachykinins are produced by endocrine cells of the gut (*Veenstra et al., 2008*; *Reiher et al., 2011*; *Lenz et al., 2001*). Interestingly, all have been implicated in regulating *dilp* expression and/or feeding behavior (*Nässel et al., 2013*; *Pool and Scott, 2014*), and most of them, namely allatostatin A1-4, diuretic hormone 31, and tachykinin 1, 2, 4, and 5, were cleaved by Nep4 (*Table 1*), indicating enzyme-substrate relationships. Thus, these results suggest that Nep4 activity

**Table 1.** Nep4 hydrolyzes peptides that regulate *dilp* expression or food intake.

Candidate peptides were analyzed for Nep4-specific cleavage. The individual molecular masses of full length peptides and cleavage products are depicted as the monoisotopic value. Cleavage positions and deviations from the respective theoretical masses (Δ) are shown separately. Cleaved peptides are highlighted in blue, and non-cleaved peptides are depicted in red. Superscripts indicate the studies that biochemically characterized the respective peptides ([1](**Baggerman et al., 2005**), [2](**Wegener et al., 2006**), [3](**Wegener and Gorbashov, 2008**), [4](**Predel et al., 2004**), [5](**Yew et al., 2009**)). n.d. indicates 'not detected', thus the respective sequences represent genomic data based predictions.

| Name | Sequence | Mass (Da) | Δ(Da) | Sequence of cleavage products | Mass (Da) | Δ(Da) | Cleavage position |
|---|---|---|---|---|---|---|---|
| Allatostatin A1 | VERYAFGLa[4] | 953.5 | −0.0676 | VERYAFG<br>VERYAF | 840.4<br>783.4 | −0.0893<br>−0.0898 | G/L<br>F/G |
| Allatostatin A2 | LPVYNFGLa[5] | 920.5 | −0.0205 | LPVYNFG<br>LPVYNF<br>LPVYN | 808.4<br>751.4<br>604.3 | −0.0492<br>−0.0148<br>−0.0223 | G/L<br>F/G<br>N/F |
| Allatostatin A3 | SRPYSFGLa[1, 4] | 924.5 | −0.0523 | YSFGLa | 584.3 | −0.0241 | P/Y |
| Allatostatin A4 | TTRPQPFNFGLa[1, 4, 5] | 1275.7 | −0.0629 | TTRPQPFNFG<br>TTRPQPFN<br>FNFGLa | 1163.6<br>959.5<br>595.3 | −0.0850<br>−0.0790<br>−0.0301 | G/L<br>N/F<br>P/F |
| AKH | QLTFSPDWa[1, 2, 3, 4] | 992.5 | 0.0051 | TFSPDWa<br>FSPDWa | 750.3<br>649.3 | −0.0360<br>−0.0473 | L/T<br>T/F |
| Corazonin | QTFQYSRGWTNa[1, 2, 3, 4, 5] | 1385.6 | −0.0582 | FQYSRGWTNa<br>QTFQYSRG | 1156.5<br>985.5 | −0.0319<br>−0.0743 | T/F<br>G/W |
| DH31 | TVDFGLARGYSGTQ-EAKHRMGLAAANFA-GGPa[n.d.] | 3149.5 | −0.0814 | YSGTQEAKHRMG<br>TVDFGLARG | 1363.6<br>934.5 | −0.1761<br>−0.0198 | G/Y; G/L<br>G/Y |
| Drosulfakinin 1 | FDDYGHMRFa[1, 4, 5] | 1185.5 | −0.0572 | FDDYGHMR | 1039.4 | −0.1147 | R/F |
| Drosulfakinin 2 | GGDDQFDDYGHMRFa[1, 4, 5] | 1657.7 | −0.0298 | GGDDQFDDYGHMR<br>FDDYGHMRFa | 1511.6<br>1185.5 | −0.1201<br>−0.0711 | R/F<br>Q/F |
| Leucokinin | NSVVLGKKQRFHSWGa[1, 3, 4, 5] | 1741.0 | −0.0905 | NSVVLGKKQRFHS<br>NSVVLGKKQRFH<br>NSVVLGKKQR<br>FHSWGa | 1498.3<br>1411.8<br>1127.7<br>631.3 | −0.1474<br>−0.1094<br>−0.1121<br>−0.0100 | S/W<br>H/S<br>R/F<br>R/F |
| sNPF1$_{1-11}$ | AQRSPSLRLRFa[2, 3, 4] | 1328.8 | −0.0520 | AQRSPSLRL | 1026.6 | −0.0962 | L/R |
| sNPF1$_{4-11}$/<br>sNPF2$_{12-19}$ | SPSLRLRFa[1, 2, 3, 4, 5] | 973.6 | −0.0859 | SPSLRLR<br>LRLRFa | 827.5<br>702.5 | −0.1543<br>−0.1451 | R/F<br>S/L |
| Tachykinin 1 | APTSSFIGMRa[1, 4] | 1064.5 | −0.0579 | APTSSFIG<br>FIGMRa | 778.4<br>621.3 | −0.0434<br>−0.0706 | G/M<br>S/F |
| Tachykinin 2 | APLAFVGLRa[1, 5] | 941.6 | −0.0396 | LAFVGLRa<br>APLAFVG<br>FVGLRa<br>APLAF | 773.5<br>673.4<br>589.4<br>517.3 | −0.0858<br>−0.0202<br>−0.0686<br>−0.0183 | P/L<br>G/L<br>A/F<br>F/V |
| Tachykinin 4 | APVNSFVGMRa[1, 4, 5] | 1075.6 | −0.0742 | APVNSFVG | 789.4 | −0.0314 | G/M |
| Tachykinin 5 | APNGFLGMRa[1, 5] | 960.5 | 0.0231 | FLGMRa | 621.3 | −0.0666 | G/F |
| Hugin | SVPFKPRLa[1, 2, 3, 4, 5] | 941.6 | −0.0776 | | | | |
| NPF | SNSRPPRKNDVNTMA-DAYKFLQDLDTYYGD-RARVRFa[n.d.] | 4278.2 | 0.50 | | | | |
| Proctolin | RYLPT[n.d.] | 648.4 | −0.0841 | | | | |
| sNPF3 | KPQRLRWa[5] | 981.6 | −0.05 | | | | |
| sNPF4 | KPMRLRWa[5] | 984.6 | −0.05 | | | | |
| Tachykinin 3 | APTGFTGMRa[1] | 935.5 | −0.0733 | | | | |
| Tachykinin 6 | AALSDSYDLRGKQQR-FADFNSKFVAVRa[n.d.] | 3087.6 | −0.1694 | | | | |

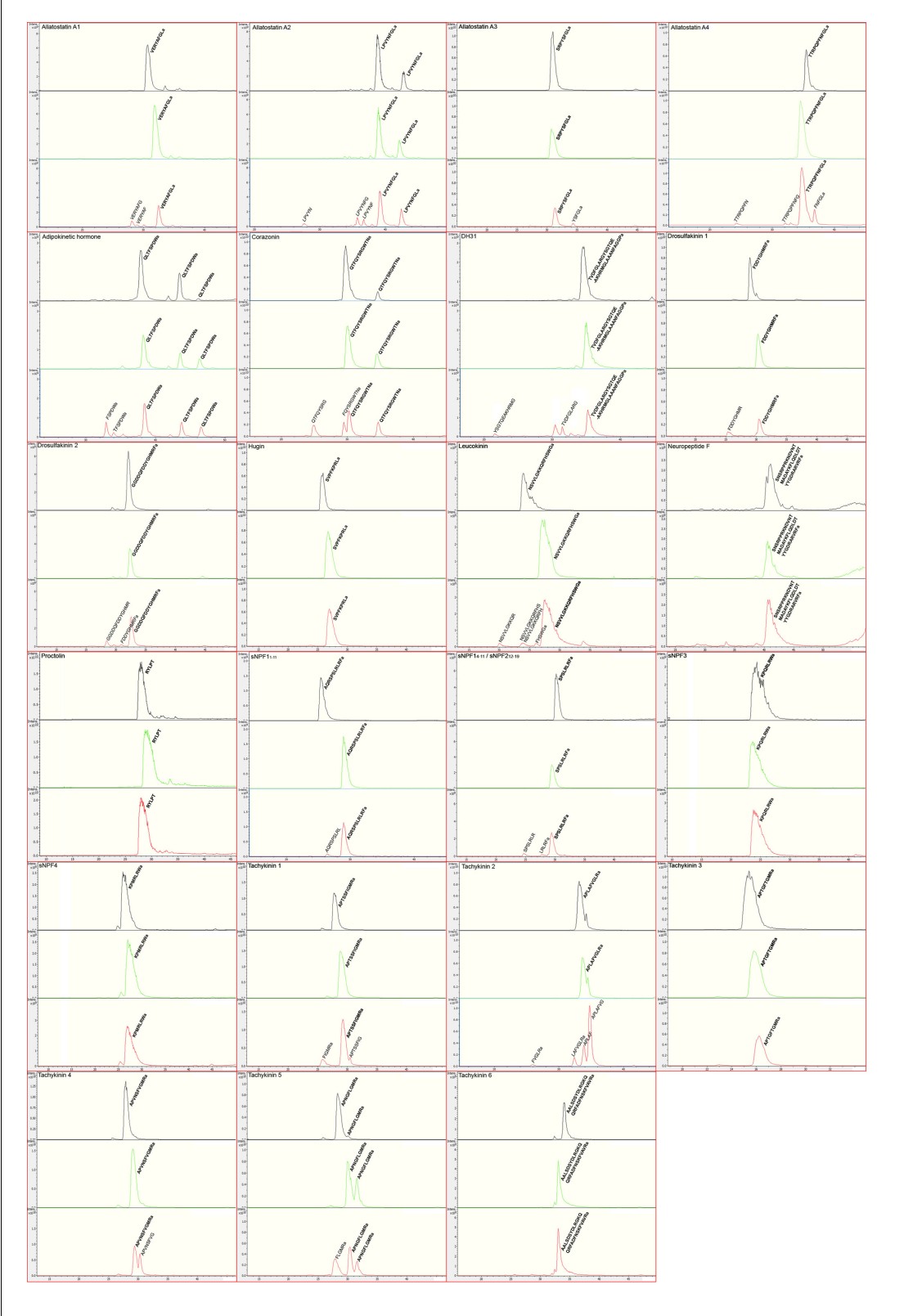

**Figure 7.** Nep4 catalyzes the hydrolysis of peptides that regulate *dilp* expression or feeding behavior. Base peak all MS chromatograms of analyzed peptides. The respective sequences of unprocessed full-length peptides (bold) and of identified Nep4-specific cleavage products are indicated. Unlabeled peaks were not identified. Spectra corresponding to untreated peptides are indicated in black, spectra corresponding to peptides incubated

*Figure 7 continued on next page*

*Figure 7 continued*

with control preparations lacking Nep4B are indicated in green, and spectra corresponding to peptides incubated with Nep4B-containing preparations are indicated in red. X-axes depict retention time (min).

The following figure supplement is available for figure 7:

**Figure supplement 1.** Heterologously expressed Nep4B can be purified to homogeneity.

at the surface of muscle cells is necessary to maintain homeostasis of distinct hemolymph circulating signaling peptides, probably gut-derived, thereby ensuring proper midgut-IPC communication. On the other hand, fat body-IPC feedback may be affected as well. However, the only factors known to mediate this process, Unpaired 2 (*Rajan and Perrimon, 2012*), DILP6 (*Bai et al., 2012*), and Stunted (*Delanoue et al., 2016* ) have molecular masses of more than 5 kDa, and thus exceed the maximum mass of a putative neprilysin substrate (*Oefner et al., 2000*). Consequently, a direct regulatory influence of Nep4 on Unpaired 2, DILP6, or Stunted activity appears unlikely.

In addition to body wall muscles, *nep4* is expressed in numerous cells of the central nervous system, predominantly in glial cells (*Figure 5*). Interestingly, compared to the muscle-specific effects, modulating *nep4* expression in this tissue has distinct and less severe effects on *dilp* expression (*Figure 3B*, *Figure 3—figure supplement 1*). This result suggests that CNS intrinsic Nep4 activity affects different neuropeptide regulatory systems than the corresponding muscle-bound activity. Considering the rather broad expression in glial cells, it is furthermore likely that the CNS regulation affects more than one system. However, localization at the IPC surface (*Figure 6*) clearly supports a direct function in the regulation of *dilp* expression. In this context, spatial proximity of the peptidase may be necessary to ensure low ligand concentrations and thus tight regulation of specific neuropeptide receptors present at the surface of IPCs. Such receptors include an allatostatin A receptor (Dar-2) (*Hentze et al., 2015*), a tachykinin receptor (DTKR) (*Birse et al., 2011*), and the short neuropeptide F receptor (sNPFR) (*Lee et al., 2008*). All are essential to proper *dilp* expression (*Lee et al., 2008*; *Hentze et al., 2015*; *Birse et al., 2011*). Interestingly, with respect to sNPFR, corresponding ligands (sNPF1$_{1-11}$, sNPF1$_{4-11}$, and sNPF2$_{12-19}$) exhibit very high-binding affinities, with IC$_{50}$ values in the low nanomolar range (*Garczynski et al., 2006*), a finding that further emphasizes the need for effective ligand clearance mechanisms in order to prevent inadvertent receptor activation. Localization of Nep4 to the surface of IPCs (*Figure 6*) and confirmation of Dar-2, DTKR, and sNPFR ligands as substrates of the peptidase (*Table 1*, *Figure 7*) strongly indicate that Nep4 participates in such clearance mechanisms.

Of note, sNPF species were detected in both, CNS and hemolymph preparations, with neuroendocrine functions of the respective peptides being suggested (*Veenstra et al., 2008*; *Garczynski et al., 2006*; *Baggerman et al., 2005*; *Wegener and Gorbashov, 2008*; *Wegener et al., 2006*). The dual localization is interesting because both compartments are accessible to Nep4, either to the CNS resident or to the muscle-bound enzyme. Significantly, sNPF is a potent regulator of *dilp* expression. Increased sNPF levels result in upregulation of *dilp* expression, and decreased sNPF levels have the opposite effect (*Lee et al., 2008*). The fact that these results inversely correlate with the effects of modulating *nep4* expression (*Figure 3*) suggests a functional relationship between sNPF and the neprilysin. Nep4-mediated cleavage of distinct sNPF species (*Table 1*, *Figure 7*) represents further evidence for this relationship.

Besides sNPF, Nep4 also cleaves corazonin, drosulfakinins, and allatostatin A (*Table 1*, *Figure 7*). Interestingly, corazonin promotes food intake (*Hergarden et al., 2012*), while allatostatin A and drosulfakinins inhibit it (*Hergarden et al., 2012*; *Söderberg et al., 2012*; *Chen et al., 2016*). This regulatory activity on peptides with opposing physiological functions indicates that Nep4 affects multiple aspects of feeding control, rather than promoting or inhibiting food intake in a mutually exclusive manner. Our finding that both, *nep4* knockdown and overexpression larvae exhibit reduced food intake (*Figure 3A*) supports this indication since it suggests that regular Nep4 activity adjusts the general peptide homeostasis in a manner that promotes optimal food intake, with deviations in either direction being deteriorative. The result that *nep4* knockdown animals exhibit reduced food intake for only up to 20 min of feeding (*Figure 3A*) may reflect this complex regulation since it indicates that at the onset of feeding reduced cleavage of peptides inhibiting food intake (e.g.

allatostatin A, drosulfakinins) is a dominant factor. With ongoing feeding, accumulation of peptides promoting food intake (e.g. corazonin) may become decisive, thus restoring intake rates.

In addition, Nep4 hydrolyzes numerous peptides that regulate *dilp* expression, including tachykinins, allatostatin A, and sNPF. However, AKH, a functional homolog of vertebrate glucagon that acts antagonistically to insulin, is also a substrate of Nep4 (*Table 1*, *Figure 7*). This finding indicates that the Nep4-mediated regulation of *dilp* expression and sugar homeostasis can also not be attributed to a single substrate or cleavage event. Rather, it is a result of the concerted hydrolysis of several critical peptides, including both, hemolymph circulating and CNS intrinsic factors. Taking into account that overexpression and knockdown of *nep4* have discrete effects on *dilp* expression (*Figure 3B*), but comparable effects on feeding (*Figure 3A*), it furthermore appears likely that dysregulation of the Nep4-mediated peptide homeostasis affects both processes somewhat independently of each other. The fact that among the novel Nep4 substrates we identified peptides that presumably affect either *dilp* signaling (e.g. DH$_{31}$), or food intake (e.g. leucokinin, drosulfakinins) in a largely exclusive manner supports this indication.

Because neprilysins and many of the novel substrates identified in this study are evolutionarily conserved factors, neprilysin-mediated regulation of insulin-like peptide expression and feeding behavior may be relevant not only to the energy metabolism in *Drosophila*, but also to corresponding processes in vertebrates, including humans. Interestingly, a critical function of murine Neprilysin in determining body mass has already been reported. The regulation depended primarily on the catalytic activity of peripheral NEP, while the CNS-bound enzyme was less important (*Becker et al., 2010*). However, until now, the underlying physiology has been obscure, essentially because no causative hydrolysis event had been identified. Our finding that also in *Drosophila* mainly peripheral (muscle-bound) Nep4 activity affected body mass, while CNS-specific modulations had only minor effects on size or weight (*Figure 1*), indicates that the neprilysin-mediated regulation of food intake, body size and insulin expression involves similar physiological pathways in both species. Furthermore, the fact that altered catalytic activity and thus abnormal peptide hydrolysis is a critical factor in mice (*Becker et al., 2010*) and in *Drosophila* (*Figures 1* and *3*) emphasizes the need to generate comprehensive, enzyme-specific lists of neprilysin *in vivo* substrates. In this context, the results of our screen for novel Nep4 substrates (*Table 1*, *Figure 7*) may be a valuable resource in order to identify corresponding substrates in vertebrates and humans.

## Materials and methods

### Fly strains

The following *Drosophila* lines were used in this work. Strain w1118 (RRID:BDSC_5905) was considered wild type. The driver lines were *mef2*-Gal4 (RRID:BDSC_27390), *repo*-Gal4 (RRID:BDSC_7415), *elav*-Gal4 (RRID:BDSC_8760), and *dilp2*-Gal4 (RRID:BDSC_37516). UAS-lines were UAS-mCherry.NLS (RRID:BDSC_38424) and UAS-2xEGFP (RRID:BDSC_6874). The *nep4*-nGFP reporter line was described previously (*Meyer et al., 2009*). *nep4* knockdown was achieved using line 100189 (KK library, no off-targets, Vienna *Drosophila* Resource Center, VDRC). A high knockdown efficiency of the respective construct was shown previously (*Panz et al., 2012*). To confirm specificity of the knockdown, a line being homozygous for both, the UAS-*nep4* RNAi construct (chromosome II) and the UAS-Nep4A overexpression construct (chromosome III) was generated and crossed to either *mef2*-Gal4 or *repo*-Gal4. Tissue-specific rescue of the respective RNAi phenotypes by simultaneous overexpression of Nep4A was used as readout for knockdown specificity. A second *nep4* RNAi construct (line 16669, GD library, VDRC) did not significantly reduce *nep4* transcript levels (*Panz et al., 2012*). It was therefore excluded from further analysis.

### Size and weight measurements

Staged (AEL 74–78 hr) male third instar larvae where grouped into genotype-specific cohorts of 10 individuals. The weights of at least five cohorts per genotype were averaged to calculate the mean weight of one respective larva. For size measurements, larvae where exposed to 60℃ water for 10 s, resulting in maximum relaxation of the body. Subsequently, animals where photographed on scale paper using a stereomicroscope (Leica MZ16 FA), and individual lengths were calculated with the Adobe Photoshop CS5 measure tool using the scale paper as a reference.

## Lethality assay

Animals of different genotypes were raised at 27 °C on apple agar plates supplemented with excess yeast paste. Stage-specific lethality rates were determined by calculating the percentage of animals of a specific stage that did not develop into the next stage. For each genotype and biological replicate, 550 embryos were analyzed. Three independent biological replicates were conducted.

## Feeding assay

Staged (AEL 74–78 hr) male third instar larvae were starved for 1 hr. Subsequently, animals were fed with dyed yeast (0.3 mg Carmin, 4 mg dry yeast, dissolved in 10 ml $H_2O$) for 5, 10, 20, or 40 min, respectively, washed, and photographed (Stemi 2000-C, Zeiss, Jena, Germany). Dye intensities (no. of detected pixels) within the intestines were determined with Fiji software (http://fiji.sc/). At least six individuals per genotype and time point were analyzed.

## NMR metabolomics

Staged (AEL 74–78 hr) male third instar larvae where grouped into genotype-specific cohorts, and six cohorts per genotype were independently analyzed to assess metabolite composition. Briefly, animals (50 mg/cohort) were homogenized (glass-Teflon homogenizer) in 500 µl ice-cold $ACN/H_2O$ (50%) and centrifuged (10,000 × $g$, 10 min) to remove fly debris and precipitate. The resulting supernatant was lyophilized and frozen at −80°C for later use. Samples were rehydrated in 650 µl of 50 mM phosphate buffer in $D_2O$ (pH 7.4) containing 50 mg/l 3-trimethylsilyl propionic acid $D_4$ (TSP) as a chemical shift reference and 50 mg/l sodium azide to prevent bacterial growth. The NMR measurements were carried out at 25 °C on a Bruker Avance-III 600 spectrometer (Bruker Biospin, Germany) equipped with a double tuned $^1H$-$^{13}C$ 5 mm cryoprobe and operated at a $^1H$ frequency of 600.13 MHz. The $^1H$ NMR spectra were acquired using a single 90° pulse experiment with a Carr Purcell Meiboom Gill (CPMG) delay added, in order to attenuate broad signals from high molecular weight components. The total CPMG delay was 40 ms, and the spin echo delay was 200 µs. The water signal was suppressed by pre-saturation of the water peak during the relaxation delay of 4 s. A total of 96k data points spanning a spectral width of 20 ppm were collected in 128 transients. For assignment purposes, two-dimensional $^1H$-$^1H$ TOCSY and $^1H$-$^{13}C$ HSQC spectra were acquired. The spectra were processed using iNMR (www.inmr.net). An exponential line broadening of 0.5 Hz was applied to the free induction decay, prior to Fourier transformation. All spectra were referenced to the TSP signal at −0.017 ppm, automatically phased and baseline corrected. The spectra were aligned using Icoshift (*Savorani et al., 2010*), and the region around the residual water signal (4.88–4.67 ppm) was removed. The integrals were normalized to total weight, and the data were scaled using pareto scaling (*Craig et al., 2006*) and centered.

## NMR data analysis

Initially, the whole dataset was subjected to principal component analysis (PCA) (*Stoyanova and Brown, 2001*). Afterwards, orthogonal projection to latent structures discriminant analysis (OPLS-DA) models were created to separate either larvae overexpressing *nep4* from control larvae or *nep4* knockdowns from control larvae. OPLS-DA models are multivariate models that predict group membership based on a multivariate input, in this case the NMR spectra. The model separates variations due to group membership from other (orthogonal) variations (*Bylesjö et al., 2006*). The OPLS-DA models were validated by cross validation where models were made with randomly chosen groups of samples left out one at a time, and group membership was predicted for the left out samples. The predictability ($Q^2$) of the models, i.e. the correlation between predicted and actual classification, was 0.95 for the comparison between *mef2*-Gal4 x w1118 and *mef2*-Gal4 x UAS-Nep4A, and 0.74 for the comparison between *mef2*-Gal4 x w1118 and *mef2*-Gal4 x UAS *nep4*-RNAi, respectively, indicating high-quality models. The loadings and the correlation coefficient (R) between intensities at the individual frequencies and the predictive component were calculated. A cutoff value for $R^2$ corresponding to p<0.05 with Bonferroni correction for an assumed number of 100 metabolites was calculated from the distribution of $R^2$ values in 10,000 permutated data sets. Signal assignments were based on chemical shifts, using earlier assignments and spectral databases described elsewhere (*Cui et al., 2008*; *Malmendal et al., 2006*; *Pedersen et al., 2008*). All multivariate analysis was performed using the Simca-P software (Umetrics, Sweden).

## Cell culture and enzymatic cleavage assay

Heterologous expression was performed in SF21 cells (RRID:CVCL_0518) using the Bac-to-Bac baculovirus expression system (Life Technologies, Carlsbad, CA, USA). The *nep4B* coding sequence was fused to a C-terminal His-tag using appropriate primer design and cloned downstream of the polyhedrin promoter into an *E.coli/S.cerevisiae*/Baculovirus triple-shuttle derivative of the pFastBac Dual vector adapted for cloning by homologous recombination *in vivo*. The respective vector (pJJH1460) was constructed similar to the vectors described in (*Paululat and Heinisch, 2012*). To track transfection efficiency, an *egfp* reporter gene was inserted into the same vector under the control of the p10 promoter. Transfected and non-transfected SF21 cells were cultured in 75-cm$^2$ flasks for 72 hr and harvested by centrifugation (300 × *g*, 5 min). Subsequently, cells were resuspended in 5 ml binding buffer (50 mM NaH$_2$PO$_4$, pH 7.9; 300 mM NaCl) and lysed with a glass-Teflon homogenizer. The resulting homogenates were centrifuged (10 min, 10,000 × *g*), and the supernatants were subjected to gravity-flow-based His-tag purification according to the manufacturer's instructions (Protino Ni-NTA agarose, Macherey-Nagel, Düren, Germany). To measure enzymatic activity, 2.5 µl of Nep4B-containing (10 ng/µl, purified from *nep4B* transfected cells) and non-containing (from untransfected control cells) preparations were supplemented with 3.5 µl (150 ng) of individual peptides. After 5 hr of incubation (35°C), 1 µl of each respective preparation was analyzed with ESI mass spectrometry. Peptides were synthesized at JPT Peptide Technologies (Berlin, Germany) with more than 90% purity. Individual cleavage assays were repeated at least three times.

## Mass spectrometry

Samples were loaded onto a trap column (Acclaim PepMap C18, 5 µm, 0.1 × 20 mm, Thermo Scientific, Sunnyvale, CA, USA) and washed. The trap column was switched inline with a separation column (Acclaim PepMap C18 2 µm, 0.075 × 150 mm, Thermo Scientific). Subsequently, bound substances were eluted by changing the mixture of buffer A (99% water, 1% acetonitrile, 0.1% formic acid) and buffer B (80% acetonitrile, 20% water and 0.1% formic acid) from 100:0 to 20:80 within 45 min. The flow rate was kept constant at 0.3 µl/min. Successively eluted compounds were analyzed with an ESI-ion trap (Amazon ETD Speed with a captive spray ionization unit, Bruker Corporation, Billerica, MA, USA) by measuring the masses of the intact molecules as well as the masses of the fragments, which were generated by collision-induced dissociation (CID) of the corresponding parent ion.

All acquired data were used for determination of peptide-specific amino acid sequences with the Mascot search algorithm (Matrix Science, Boston, MA, USA) in combination with a custom-made database containing 37 different sequences of peptides. To avoid an increased false-positive identification rate the p-value was lowered to 0.005 (resulting in an individual ion score > 18). As enzyme, the option 'none' was chosen. Thus, every subsequence of every protein was used for identification.

## Immunohistochemistry

Brains prepared from staged male third instar larvae (AEL 74–78 hr) were fixed (3.7% formaldehyde, 1 hr) and permeabilized (1% Triton X-100, 1 hr). Subsequently, tissues were incubated in PBS containing 0.15% SDS (30 min), blocked with Roti-Block (Carl Roth, Karlsruhe, Germany) for 45 min, washed in PBT (4×, 10 min each), and incubated in Roti-Block (45 min) and primary antibody (overnight). Samples were washed in PBT (4×, 10 min each) and blocked again as described above. Secondary antibodies were applied simultaneously for 90 min. Finally, samples were washed as described above and mounted in Fluoromount-G (SouthernBiotech, Birmingham, USA). For staining of body wall muscles, male third instar larvae were dissected on Sylgard plates (Sylgard 184 Elastomer Base and Curing Agent, Dow Corning, Michigan, USA), fixed in 3.7% formaldehyde in PBS for 1 hr, rinsed three times in PBS, and transferred into 1.5 ml reaction cups. Subsequently, tissues were permeabilized in 1% Triton X-100 for 1 hr, blocked in Roti-Block (45 min), and incubated with primary antibodies (overnight). Samples were washed in PBT (3×, 10 min each) and blocked again as described above. Secondary antibodies were applied for 90 min. Finally, samples were washed as described above and mounted in Fluoromount-G (SouthernBiotech, Birmingham, USA). The primary antibodies used were: anti-Nep4 (RRID:AB_2569115, 1:200, raised in rabbit, monospecificity was confirmed in [*Meyer et al., 2009*]), anti-GFP (RRID:AB_889471, 1:500, raised in mouse), anti-GFP (RRID:AB_305564, 1:2000, raised in rabbit), anti-HA (RRID:AB_262051, 1:100, raised in mouse), and

anti-Repo (RRID:AB_528448, 1:5, raised in mouse). The secondary antibodies were anti-mouse-Cy2 (RRID:AB_2307343, 1:100, raised in goat), anti-mouse-Cy3 (RRID:AB_2338680, 1:200, raised in goat), anti-rabbit-Cy2 (RRID:AB_2338021, 1:100, raised in goat), and anti-rabbit-Cy3 (RRID:AB_2338000, 1:200, raised in goat). Confocal images were captured with an LSM5 Pascal confocal microscope (Zeiss, Jena, Germany). To exclude a possible bleed-through of the signals, sequential channel acquisition was performed starting with Cy3 channel by using single excitation at 543 nm and a long pass emission filter LP560, followed by Cy2 channel acquisition with single excitation at 488 nm and a single bandpass filter BP 505–530 nm. There was no bleed-through of the Cy2 signal to the Cy3 channel because Cy2 is not excited by the 543 nm laser line. Using a narrow bandpass filter between 505 nm and 530 nm guaranteed that cross talk of Cy3 excitation by the 488 laser line is not detected during Cy2 channel acquisition. Z-stacks are displayed as maximum projections if not stated otherwise.

## qRT-PCR

Total-RNA (RNeasy Mini Kit, Qiagen, Hilden, Germany) from staged male third instar larvae (AEL 74–78 hr) was treated with DNase I (Invitrogen, Carlsbad, CA, USA) according to the manufacturer's instructions and used as a template for cDNA synthesis (AMV First Strand cDNA Synthesis Kit for RT-PCR, Roche). qRT-PCR was conducted according to standard protocols using DyNAmo ColorFlash SYBR Green qPCR Kit (Biozym, Hessisch Oldendorf, Germany) and an iCycler iQ Real-Time PCR System (Bio-Rad, Munich, Germany). Data were evaluated as described in (*Simon, 2003*). All experiments were repeated at least three times (individual biological replicates, each consisting of at least three technical replicates). The sequences of primers used were as follows: *dilp1*, 5´-GGGGCAGGA TACTCTTTTAG-3´ and 5´-TCGGTAGACAGTAGATGGCT-3´; *dilp2*, 5´-GTATGGTGTGCGAGGAGTA T-3´ and 5´-TGAGTACACCCCCAAGATAG-3´; *dilp3*, 5´-AAGCTCTGTGTGTATGGCTT-3´ and 5´-AGCACAATATCTCAGCACCT-3´; *dilp5*, 5´-AGTTCTCCTGTTCCTGATCC-3´ and 5´-CAGTGAGTTCA TGTGGTGAG-3´; *rp49*, 5´-AGGGTATCGACAACAGAGTG-3´ and 5´-CACCAGGAACTTCTTGAATC-3´.

## Statistics

Statistical analysis (one-way ANOVA with pairwise comparisons) was performed using OriginPro 8 software (OriginLab Corporation, Northampton, MA, USA).

## Acknowledgements

We thank Mechthild Krabusch and Martina Biedermann for excellent technical assistance. We also thank the Bloomington *Drosophila* Stock Center and the Vienna *Drosophila* Resource Center for providing fly stocks, and Christian Wegener for sharing peptides. We gratefully acknowledge also Flemming Hofmann Larsen and Søren Balling Engelsen (University of Copenhagen, Department of Food Science) for the use of their 600 MHz spectrometer. This work was supported by the 'Incentive Award of the Faculty of Biology/Chemistry' (University of Osnabruck) to HM, by grants from the German Research Foundation to AP and HM (SFB 944: Physiology and dynamics of cellular microcompartments), and by a grant from the FAZIT foundation to BH. AP received additional funding from the State of Lower-Saxony, Hannover, Germany (11-76251-99-15/12 (ZN2832)).

## Additional information

### Funding

| Funder | Grant reference number | Author |
| --- | --- | --- |
| Deutsche Forschungsgemeinschaft | SFB944 | Achim Paululat<br>Heiko Meyer |
| University of Osnabrück | Incentive Award | Heiko Meyer |
| FAZIT Stiftung | PhD stipend | Benjamin Hallier |
| State of Lower Saxony | 11-76251-99-15/12 (ZN2832) | Achim Paululat |

The funders had no role in study design, data collection and interpretation, or the decision to submit the work for publication.

## Author contributions

BH, RS, SW, AM, Acquisition of data, Analysis and interpretation of data, Drafting or revising the article; EC, JV-F, Acquisition of data, Drafting or revising the article; JJH, AP, Analysis and interpretation of data, Drafting or revising the article; HM, Conception and design, Acquisition of data, Analysis and interpretation of data, Drafting or revising the article

## Author ORCIDs

Heiko Meyer, http://orcid.org/0000-0002-3304-4523

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
