## [Decision Letter]

Thank you for submitting your article "*Drosophila* neprilysins control insulin signaling and food intake via cleavage of regulatory peptides" for consideration by *eLife*. Your article has been reviewed by three peer reviewers, and the evaluation has been overseen by a Reviewing Editor and K VijayRaghavan as the Senior Editor. The reviewers have opted to remain anonymous.

The reviewers have discussed the reviews with one another and the Reviewing Editor has drafted this decision to help you prepare a revised submission.

Summary:

This manuscript examines the role of neprilysins in regulating *Drosophila* behavior and physiology. They show that overexpression and knock down of Nep4 in muscle tissue cause stage specific lethality and size/mass difference in the 3rd instar larvae. Manipulation of Nep4 also potently impacts development and the metabolome. The authors also assessed the endogenous expression of Nep4 in larval tissue and find that Nep4 is expressed in the muscle as well as in the central nervous system, including in the neuroendocrine cells that produce insulin like peptides. Given the role of neprilysins in regulation of neuropeptide signaling and the novelty of these phenotypes, these findings are interesting and could be of interest to a broad audience. However, currently the data presented is somewhat preliminary and several issues need to be addressed. In particular, the authors need to clarify how Nep4 is regulating metabolism, through muscle or through other cell types including neurons.

Essential revisions:

1) In this manuscript, the authors use *mef2*-Gal4 to knock down Nep4 expression in muscle tissue. They conclude that Nep4 acts on the muscle tissue to regulate metabolism. However, the authors also show that Nep4 is expressed in the nervous system including the cells that produce the insulin like peptides. Although *mef2* is a well-known muscle marker in *Drosophila, mef2* gene and enhancers are also expressed in other types of cells, including neurons. These neurons include clock cells, which may play a direct role in regulating insulin signaling, release and therefore larval metabolism and feeding behavior.

The authors could asses the expression pattern of the *mef2*-Gal4 driver that is used in the manuscript to exclude the possibility that the phenotypes observed in Figure 1–Figure 3 are not neuronal but muscle specific.

The authors could also try a neuron specific driver such as *elav*-Gal4 or *nsyb*-Gal4 to repeat the experiments in Figure 1. If the Nep4 function is required in the muscle then neuron specific drivers will have no effects on lethality, body size or mass.

2) Another major concern is the specificity of the RNAi experiments. In some cases there is little effect of RNAi, but a large effect with overexpression, while in other situations the opposing manipulations result in the same phenotype. It is easy to imagine that the effect of Nep4A is resulting in non-physiological effects? This is particularly important for interpreting the phenotypes of reduced *dilp* expression. Additionally, is it concerning that the RNAi and overexpression have comparable effects on feeding, but different effects on ilp levels?

Another issue is that the genetic control for the experiments shown in Figure 1–Figure 3 do not include a UAS-control (UAS-nep4A x w1118, UAS-*nep4* RNAi). This is an important control, as a background expression of UAS-constructs is not rare and may have effects on its own (unlike GAL4 background expression). To demonstrate a lack of effect of the UAS construct alone, however, at least data for the effects of the UAS-controls in the genetic experiments shown in Figure 1 should be included.

3) Another issue is whether the authors need to include heterozygote controls. If there is a small ubiquitous background expression of Nep4, it is not difficult to see that this might have effects by itself.

---

## [Author Response]

*[…] Essential revisions:*

*1) In this manuscript, the authors use mef2-Gal4 to knock down Nep4 expression in muscle tissue. They conclude that Nep4 acts on the muscle tissue to regulate metabolism. However, the authors also show that Nep4 is expressed in the nervous system including the cells that produce the insulin like peptides. Although mef2 is a well-known muscle marker in Drosophila, mef2 gene and enhancers are also expressed in other types of cells, including neurons. These neurons include clock cells, which may play a direct role in regulating insulin signaling, release and therefore larval metabolism and feeding behavior.*

*The authors could asses the expression pattern of the mef2-Gal4 driver that is used in the manuscript to exclude the possibility that the phenotypes observed in Figure 1–Figure 3 are not neuronal but muscle specific.*

*The authors could also try a neuron specific driver such as elav-Gal4 or nsyb-Gal4 to repeat the experiments in Figure 1. If the Nep4 function is required in the muscle then neuron specific drivers will have no effects on lethality, body size or mass.*

As suggested, we repeated the experiments depicted in Figure 1 using *elav*-Gal4 as a neuron-specific driver. In this novel line of experiments neither overexpression nor knockdown of *nep4* had any significant influence on viability, body size or mass. This result indicates that the effects observed upon *mef2*-Gal4 driven overexpression or knockdown are muscle-specific. The “Results” section and Figure 1 were amended accordingly.

*2) Another major concern is the specificity of the RNAi experiments. In some cases there is little effect of RNAi, but a large effect with overexpression, while in other situations the opposing manipulations result in the same phenotype. It is easy to imagine that the effect of Nep4A is resulting in non-physiological effects? This is particularly important for interpreting the phenotypes of reduced dilp expression. Additionally, is it concerning that the RNAi and overexpression have comparable effects on feeding, but different effects on ilp levels?*

To address this issue we generated flies being transgenic for both, the *nep4* RNAi construct as well as the Nep4A overexpression construct. Simultaneous overexpression of Nep4A completely rescued the *nep4* RNAi phenotypes (embryonic/pupal lethality, reduced size/weight), thus confirming specificity of the knockdown. The result that respective animals exhibited a marginally, yet significantly increased lethality rate during third instar larval stage indicates that overexpression of Nep4A is somewhat more effective than knockdown, eventually resulting in slightly increased expression levels of the peptidase. The fact that increased Nep4A levels result in elevated larval lethality was already depicted in Figure 1. The “Materials and methods”, “Results”, and Figure 1 were amended accordingly.

Regarding the observation that *nep4* RNAi and overexpression have comparable effects on feeding, but different effects on ilp levels, we do not consider this point unexpected. It rather indicates that impaired Nep4 activity, which likely causes aberrant hydrolysis of numerous peptides (Table 1), affectsboth processes somewhat independently of each other. This indication is supported by the fact that among the novel Nep4 substrates we identified peptides that presumably affect either *dilp* signaling (e.g. DH_31_), or food intake (e.g. leucokinin, drosulfakinins) in a largely exclusive manner. We amended the “Discussion” to include this point.

*Another issue is that the genetic control for the experiments shown in Figure 1–Figure 3 do not include a UAS-control (UAS-nep4A x w1118, UAS-nep4 RNAi). This is an important control, as a background expression of UAS-constructs is not rare and may have effects on its own (unlike GAL4 background expression). To demonstrate a lack of effect of the UAS construct alone, however, at least data for the effects of the UAS-controls in the genetic experiments shown in Figure 1 should be included.*

*3) Another issue is whether the authors need to include heterozygote controls. If there is a small ubiquitous background expression of Nep4, it is not difficult to see that this might have effects by itself.*

We repeated the experiments depicted in Figure 1 with the respective UAS-controls and did not observe any significant effects on viability, body size or mass. Thus, if there is a small background activity of the UAS-constructs, it is of no distinct physiological relevance. The respective UAS-controls were included in Figure 1.